# The Positive Association between Grit and Mental Toughness, Enhanced by a Minimum of 75 Minutes of Moderate-to-Vigorous Physical Activity, among US Students

**Andreas Stamatis** [1,2,*] **, Grant B. Morgan** [3] **, Ali Boolani** [4] **and Zacharias Papadakis** [5]

[1] Department of Health & Sport Sciences, University of Louisville, Louisville, KY 40292, USA
[2] Sports Medicine Institute, University of Louisville Health, Louisville, KY 40208, USA
[3] Department of Educational Psychology, Baylor University, Waco, TX 97304, USA; grant_morgan@baylor.edu
[4] Honors Program, Clarkson University, Potsdam, NY 13699, USA; ali.boolani@gmail.com
[5] Department of Health Promotion and Clinical Practice, Barry University, Miami Shores, FL 33161, USA; zpapadakis@barry.edu
[*] Correspondence: coach_stam@rocketmail.com; Tel.: +1-5028520547

**Abstract:** Drawing from the 2015 Gucciardi et al.'s mental toughness (MT) framework, this study examines the association between grit and MT in US college students, while considering the moderating role of at least 75 min of moderate-to-vigorous physical activity (MVPA) based on recommendations from the American College of Sports Medicine. We administered the Grit-S Scale and the Mental Toughness Index in two samples of a total of 340 US undergraduate student-athletes and graduate students. The Quality Assessment Tool for Observational Cohort and Cross-Sectional Studies was employed to ensure internal validity, while statistical procedures including principal component analysis and regression models were utilized to analyze the collected data. A weighted component combining grit and the interaction between MVPA and grit significantly predicted MT, explaining 23% of its variability. Drawing from a specific conceptual framework, this study provides novel insights into the relationship between grit, engagement in at least 75 min of MVPA per week, and MT among US collegiate students. The findings support a positive association between grit, MVPA, and both MT and a specific component of MT, highlighting the significance of these factors in enhancing performance and suggesting potential implications for future research and practical applications in the field.

**Keywords:** mentally tough; positive psychology; MVPA; ACSM; Gucciardi; optimal functioning

## 1. Introduction

Recent years have seen a growing interest in the concept of *mental toughness* (MT) within the scholarly community [1]. MT is "a state-like psychological resource that is purposeful, flexible, and efficient in nature for the enactment and maintenance of goal-directed pursuits" ([2], p. 18).

In their seminal MT piece, Gucciardi and colleagues [3] addressed several at-the-time conceptual and methodological concerns by conducting a series of five studies across a range of achievement contexts, including sport and education. The authors investigated the number of distinct factors that make up MT, its relationships with other psychological constructs, and its stability across different situations and over time. As a result, not only did they provide evidence for a unidimensional conceptualization of MT (vs. multidimensionality; e.g., [4]), but they also developed a new MT measurement tool (i.e., the Mental Toughness Index).

The Mental Toughness Index (MTI) is an eight-item instrument that demonstrated superior predictive validity in comparison to the widespread Mental Toughness Questionnaire-48 [4]. Each item measures (range: 1–7) one key MT dimension, and each key dimension

is supported by relevant theory and research (Table 1). Item 4 is about the *success mindset*, and one of the supporting theories is *grit*, signifying that MT is partially explained by grit.

**Table 1.** Key mental toughness dimensions, respective Mental Toughness Index items, and associated theory and research.

| MTI Item | Key Dimension | Supporting Theory and Research |
| --- | --- | --- |
| I believe in my ability to achieve my goals | Generalized Self-efficacy | Self-efficacy theory |
| I am able to regulate my focus when performing tasks | Attention Regulation | Cognitive control perspectives Executive functions |
| I am able to use my emotions to perform the way I want to | Emotion Regulation | Process and goal-oriented models of emotion regulation |
| I strive for continued success | Success Mindset | Grit Hope theory |
| I execute my knowledge of what is required to achieve my goals | Context Knowledge | Cognitive theories of wisdom Hope theory Performance intelligence |
| I consistently overcome adversity [a] | Overcoming Adversity | |
| I am able to execute appropriate skills or knowledge when challenged | Buoyancy | Academic and workplace buoyancy |
| I can find a positive in most situations | Optimistic Style | Optimism Explanatory style |

Note. In order to comprehensively assess a wide range of facets, a single item was preserved for each distinct key dimension (seven items), alongside an additional item designed to gauge an individual's aptitude for handling adverse circumstances (Item 6). This approach (a total of eight items) ensured the inclusion of both routine challenges and significant distressing events within the evaluation framework [3]. [a] It is important to highlight that, in divergence from the remaining items, the authors of the seminal paper did not include specific theoretical frameworks or research findings to substantiate Item 6 within *their* Table 2 [3]. Nevertheless, subsequent personal communication with the lead author yielded recommendations for pertinent sources, namely Cohen and colleagues [5] and Luhmann and colleagues [6], which were deemed valuable references for initiating further exploration of the subject matter.

Grit is a trait that has been defined as perseverance and passion for long-term goals [7]. *Perseverance* is one of the two grit factors and refers to the sustained application of effort towards a goal over time, even in the face of adversity. The second factor is *passion*, and it refers to the maintenance of interest in a particular field or pursuit over time [8].

MT and grit are regarded as *Positive Psychology* [9] constructs as they both are considered to contribute to individual optimal functioning [2,10]. Other conceptual commonalities include goal orientation [11,12] and persistence in facing adversity [13,14]. Due to the above, these two terms are occasionally used interchangeably despite their substantial differences [2].

Perhaps the most noteworthy difference can be described through the field of *Personality Psychology* [15]. In terms of patterns of cognitions, emotions, and behaviors, *traits* are enduring and stable and manifest similarly across various situations while *states* are temporary and specific to a particular moment in a given situation [16]. Grit is considered a dispositional trait [7] while MT is regarded as a state-like resource [2]. Therefore, MT consists of features that persist (similar to grit) yet fluctuate in response to situational or temporal variations and are susceptible to growth or alteration (in contrast to grit).

Despite their distinctiveness, both grit and MT have been found to be determinants of success in several performance settings, including education and exercise. In terms of education, self-reported grit scores have been found to be related to grade point average (GPA), with perseverance being a better predictor of GPA than passion [8,17,18]; similarly, the grades of college students with high MT were found to be significantly higher compared to those with low MT [19,20].

Evidence inferring the value of grit and MT has been found in physical activity/exercise/ sport settings, as well. A meta-analytic study, for example, found that grit moderated the relationship between motivational feedback and subsequent athletic performance [21], while another study identified grit as a distinguishing feature of super-elite athletes [22]. In terms of moderate- and vigorous-intensity physical activity, grit was found to be a significant predictor [23]. Concerning MT, one review suggested that MT may be a more important predictor of performance outcomes than athletes' skills or physical abilities [24], while another review recommended that MT may be beneficial to athletic success [25]. In addition, without establishing the direction of the relationship, Gerber and colleagues [26] reported a positive correlation between MT and physical activity.

*Physical activity* (PA) is defined by Caspersen and colleagues [27] as movement resulting from the contraction of skeletal muscle that leads to an increase in energy expenditure beyond the level of energy used while at *rest* (1 *Metabolic Equivalent of Task*; MET); *exercise* is considered a type of planned and goal-oriented recreational PA [28]. Recently, the American College of Sports Medicine (ACSM) has shifted its focus to *moderate-to-vigorous PA* (MVPA; [29]). That emphasis on MVPA (i.e., at least 75 min per week) in the 11th edition of the *ACSM's Guidelines for Exercise Testing and Prescription* (GETP) was justified not only because higher intensity PA is associated with additional health benefits (*dose–response relationship;* [29]) but also because Ekelund and colleagues [30] found MVPA to reduce the mortality risk associated with *sedentary behavior* ($\leq$1.5 METs while in a sitting, reclining, or lying posture; [31]).

Despite its benefits, a majority of US adults are not engaging in adequate amounts of PA [32]. In terms of US students, it has been documented that graduates reported exercising less than undergraduates [33] and that student-athletes reported engaging in more MVPA than non-athletes [34].

Except for differences in MVPA engagement (central to our study), these particular sub-populations demonstrate distinct lifestyles, too (e.g., [35,36]). For instance, undergraduate student-athletes often face the unique pressures of balancing academic responsibilities with rigorous physical training. On the other hand, graduate students present a different set of challenges and stressors associated with advanced academic and professional pursuits. The examination of the association between grit and MT, utilizing specifically Gucciardi's [3] conceptual framework (i.e., unidimensional notion of MT partially explained by grit), while also considering the potential impact of MVPA as emphasized by the ACSM, remains unexplored within this sample.

In addition, while it is conceivable that MVPA could be an outcome of grit and MT (i.e., individuals high in these constructs might be more likely to engage in PA), studies have also indicated that PA could augment psychological resources (e.g., [37–41]). Consequently, the potential of MVPA to amplify or strengthen the psychological benefits of grit and MT (i.e., MVPA being a moderator in the grit–MT relationship) is plausible.

The omission of the examination of the association of these two constructs in these sub-populations in the current literature represents a missed opportunity for potential discoveries in both basic and applied research. Therefore, based on Gucciardi's unidimensional MT conceptualization, the purpose of this exploratory/hypothesis-generating study was to investigate if grit is associated with MT in US college students and the role of MVPA in that relationship. More specifically, there were three research questions (RQs):

1. Does grit (and its subcomponents) explain the variability in MT in US collegiate students (undergraduate student-athletes vs. graduate students)?
2. Does grit (and its subcomponents) explain the variability in US collegiate students' (undergraduate student-athletes vs. graduate students) response to MTI item 4 (i.e., "I strive for continued success")?
3. Are the above relationships moderated by at least 75 min of MVPA?

## 2. Materials and Methods

Below, we present information about the sample, the implemented protocols, the utilized measurement tools, and the applied statistical procedures. Before that, we showcase the measures we proactively took to ensure the internal validity of our study.

In more detail, we utilized the Quality Assessment Tool for Observational Cohort and Cross-Sectional Studies, which was developed by the National Institutes of Health (NIH) through the National Heart, Lung, and Blood Institute in 2013 [42] (National Heart, Lung and Blood Institute, 2013). The present tool has been meticulously crafted to systematically detect prospective weaknesses in the methodologies and execution of cross-sectional studies, thereby facilitating the evaluation of a study's quality and adherence to rigorous standards of internal validity. Subsequently, in Table A1 (See Appendix A), we delineate each criterion encompassed within the tool and expound upon its application within the context of our study.

### 2.1. Participants

The sample consisted of 340 US collegiate students, of whom 280 were graduate students and 60 were undergraduate student-athletes. The distribution of reported race/ethnicity is provided in Table 2. As expected, the graduate student sample was slightly older ($M = 27.8$, $SD = 7.6$), on average, than the undergraduate student-athletes ($M = 21.1$, $SD = 2.0$). All participants in the study were carefully selected based on predetermined inclusion criteria (see Procedures) to ensure they belonged to similar populations and were recruited within the same period. No significant differences/relationships were observed on the outcome variables (discussed below) on the basis of race/ethnicity, sex, or age.

**Table 2.** Distributions of reported race/ethnicity by sample.

| Race/Ethnicity | Student-Athlete | | Graduate Student | |
|---|---|---|---|---|
| | **N** | **%** | **N** | **%** |
| African American | 2 | 3 | 12 | 4 |
| Asian/Asian American | 0 | 0 | 15 | 5 |
| Native Hawaiian or other Pacific Islander | 0 | 0 | 13 | 5 |
| White | 52 | 87 | 240 | 86 |
| Did not report | 6 | 10 | 0 | 0 |

### 2.2. Procedures

Data collection for the study involved the utilization of the Qualtrics platform to gather information pertaining to MT, MVPA, grit, and participant demographics. Electronic informed consent was obtained from all participants prior to their involvement in the study. The methodological design was cross-sectional. The data collection process commenced in February and concluded in March 2021. The distribution of the survey was primarily facilitated through the professional networks of the authors, resulting in a final sample that can be characterized as the product of the combination of convenience and purposive sampling approaches. In order to be included in the study, participants were required to meet the inclusion criteria of being either an undergraduate student-athlete or a graduate student enrolled in a higher education institution within the United States.

### 2.3. Measures

2.3.1. The Mental Toughness Index

The Mental Toughness Index (MTI) is a self-report inventory developed based on the work of Gucciardi et al. [3] to measure individual levels of MT. As shown in Table 1, the MTI consists of eight items (e.g., I strive for continued success, I consistently overcome adversity) that are scored from 1 (False, 100% of the time) to 7 (True, 100% of the time) and are weighted equally in the scoring. The total score ranges from 8 to 56, with higher scores indicating higher individual MT. Since its development, several studies have provided

reliability and validity evidence supporting the use of the MTI to make inferences from the scores [43–46].

### 2.3.2. The Grit-S Scale

The Grit-S Scale [8] is a shorter version of the original psychometric instrument (i.e., Grit-O Scale; 12 items) comprising 8 items designed to assess an individual's level of grit. This assessment instrument contains four items for perseverance (e.g., I am a hard worker) and four for passion (e.g., My interests change from year to year) with responses ranging from 1 (Not like me at all) to 5 (Very much like me). Overall, the Grit-S Scale stands as an instrument that provides reliable scores for measuring an individual's inclination to sustain interest in and effort toward long-term goals and making valid inferences based on them [47–49].

### 2.3.3. Moderate-to-Vigorous Physical Activity

Participants were asked to report their current weekly PA of moderate ("you can talk but not sing during the activity")/vigorous ("you will not be able to say more than a few words without pausing for a breath") intensity. This metric was reported in minutes per week.

### *2.4. Data Analysis*

To answer the RQs stated above, we engaged the following data analysis plan. First, we conducted a power analysis to determine how many observations would be necessary to yield adequate power for the statistical models. Due to the specificity and sophistication of our model, we performed the power analysis using Monte Carlo simulation methods. That is, we specified the anticipated regression model within a structural equation modeling framework and generated 500 samples of iteratively increasing sample sizes. For each sample size, we calculated the proportion of times out of 500 replications the null hypothesis was rejected (i.e., power). We repeated this process using the lavaan package in *R* until the proportion of significant findings for the effects of interest was at least 0.8. The result of the power analysis indicated a sample size of 300. We aimed to collect 10% additional observations to account for observations that may not be usable. Second, after we collected data, we conducted descriptive analyses to examine the differences between the graduate student and student-athlete samples. Third, we applied confirmatory factor analysis (CFA) to the responses from each instrument to evaluate the strength of the reliability and validity evidence. Fourth, we saved the factor scores from the CFA model output to use in subsequent analyses. The factor scores were adjusted for measurement error (i.e., lack of reliability), which enhanced the power of the statistical analyses. Finally, we estimated regression models using the factor scores for the grit subscales as predictors and MT as the outcome; we also included the interactions between the grit subscales and MVPA to examine MVPA as a possible moderator.

## 3. Results

### *3.1. Descriptive Analysis*

On average, the participants reported about 435 (*SD* = 602) minutes of MVPA per week, but there were marked differences between the respondent types. Graduate students reported about 311 (*SD* = 400) minutes per week, whereas student-athletes, as expected, reported many more minutes; that is, they reported about 1145 (*SD* = 975) minutes. This overlap between student type and MVPA had an implication for future analyses, which is discussed in more detail below.

### *3.2. Evaluation of CFA Model Solutions and Reliability Estimates*

The application of confirmatory factor analysis (CFA) in our study served multiple key purposes. Primarily, CFA was utilized to ensure that the constructs of mental toughness (MT) and grit were measured accurately within our specific collegiate student sample.

This is particularly important given that scales and measurements, while standardized, can often manifest differently across diverse populations. By conducting CFA, we were able to confirm that the dimensions of MT and grit, as theorized and operationalized in the original scales (i.e., Mental Toughness Index and Grit-S Scale), were indeed reflected in our sample. This step was crucial to validate the underlying factor structure of these constructs in our specific demographic, ensuring that subsequent analyses were based on sound and appropriate measurement models. The validity of our findings heavily relies on this process, as it confirms that the tools used are measuring the intended constructs accurately within the context of our study.

In more detail, the CFA model solutions were examined to ensure there was adequate model–data fit using Mplus (version 8.8; [50]). The grit instrument was modeled such that the passion and perseverance domains were separate, but the correlation between the two was freely estimated. The two-factor solution fit the data adequately well for this sample (CFI = 0.96, TLI = 0.93, RMSEA = 0.10, SRMR = 0.05) [51]. The reliability estimates for grit subdomains were 0.79 and 0.74 ($\omega$). The grit model is shown in Figure 1. MT was modeled as a unidimensional construct, and the one-factor solution fit the data adequately well for this sample (CFI = 0.96, TLI = 0.94, RMSEA = 0.15, SRMR = 0.04). The factor scores were saved from these analyses and used in the subsequent analyses. The reliability estimate for MT was 0.92 ($\omega$). The MT model is presented in Figure 2.

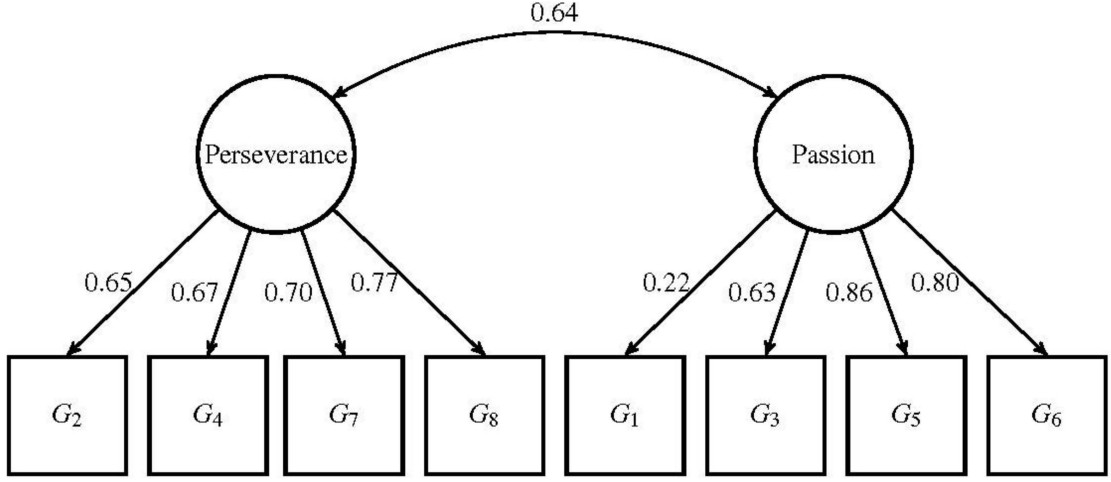

**Figure 1.** Path model showing grit, passion; grit, perseverance; and their corresponding items.

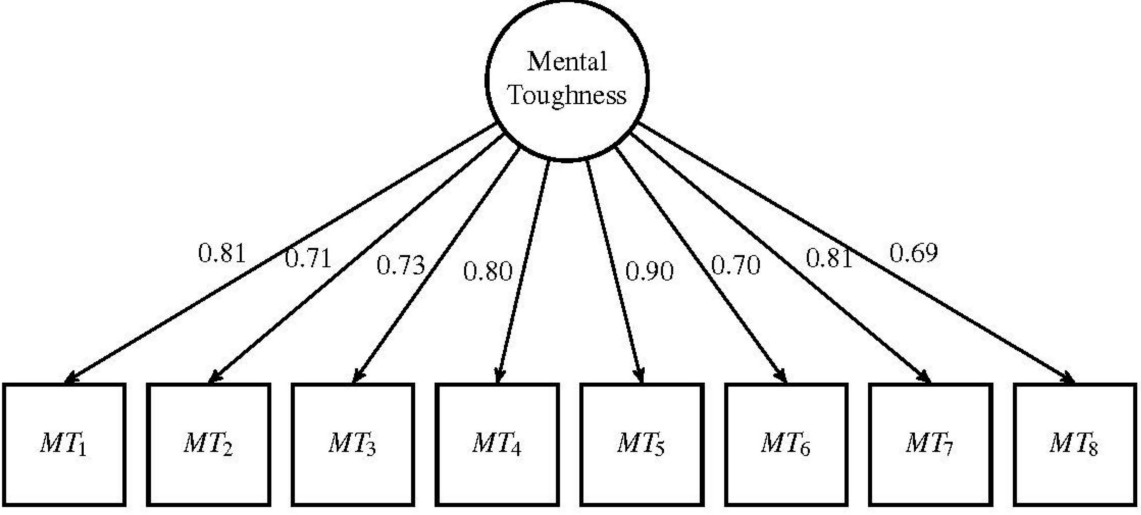

**Figure 2.** Path model showing mental toughness and its corresponding Items.

### 3.3. Blocking MVPA Engagement and Its Impact on Comparisons within Respondent Types

For analyses with the MVPA, we blocked based on whether each respondent engaged in at least 75 min of MVPA per week. Unsurprisingly, there were no student-athletes who reported engaging in fewer than 75 min of MVPA per week. As a result of the relationship between MVPA and respondent type, it was not possible to compare the effect of the blocked MVPA variable within the respondent type. In light of the focus of this investigation, the MVPA variable was used to examine the moderation effect between grit and MT.

### 3.4. Regression Model with Grit, MVPA Blocking, and MT

The factor scores for the grit subdomains and MVPA blocking variable were used in a regression model with MT as the outcome variable. The interaction between each of the grit subdomains with MVPA was included to allow MVPA to moderate the relationship between grit and MT. Although the regression was estimated, the predictor variables were very highly correlated, which violated the multicollinearity assumption for the model (i.e., variance inflation factor values ranged from 5.4 to 7.7). Due to the violation of the multicollinearity model assumption, the regression model was not interpreted. Instead, we employed a different analytic strategy to address the collinearity between predictors.

### 3.5. Principal Component Analysis and Regression Model for MT Prediction

The decision to use principal component analysis (PCA) was driven by the complexity and multidimensionality of the constructs under examination. PCA allowed us to simplify these complex relationships by reducing the data to a smaller set of summary indices (principal components) that represent the most significant patterns in the data. In our study, where multiple interrelated variables (grit subdomains, MVPA, and MT) were examined, PCA aided in distilling these variables into a more manageable form without substantial loss of information. This analytical strategy was particularly advantageous in addressing the issues of multicollinearity, which was a significant challenge in our initial regression models. By transforming the original variables into a new set of uncorrelated components, PCA enabled us to capture the essence of the data, focusing on the components that explained the majority of the variability in the relationship between grit, MVPA, and MT.

In more detail, we used PCA to create a weighted combination of the variables to explain the maximum amount of shared variability. In this case, we calculated a single weighted component that accounted for 75% of the total variability (i.e., first eigenvalue of 3.02). Within the component matrix, each of the variables was weighted heavily, but the interaction between MVPA and the perseverance grit subdomain was weighted slightly higher ($\lambda = 0.88$). The other component loadings ranged from 0.85 to 0.87. The component was then used as the predictor of MT. In the final regression model, the component consisting of grit and grit-by-MVPA interactions explained 23% of the variability in MT ($\beta = 0.48$, $p < 0.001$; Figure 3). Each of the variables was positively associated with the component, and thus, the positive sign of the regression coefficient indicates that grit was positively related to MT and that engaging in at least 75 min of MVPA further strengthened the relationship. Unfortunately, due to the high level of multicollinearity and component adjustment, we could not identify the unique contribution of these variables via the squared semi-partial correlation coefficients.

### 3.6. Analysis of Item 4 from the MTI with Grit and MVPA Component

The same set of analyses was conducted using Item 4 from the MTI as the outcome variable, and the general findings followed the same pattern but did not show the same strength. That is, there was a positive relationship between the grit and MVPA component and responses to Item 4 ($\beta = 0.35$, $p < 0.001$; Figure 4). The component also accounted for about 12% of the variability in responses to Item 4.

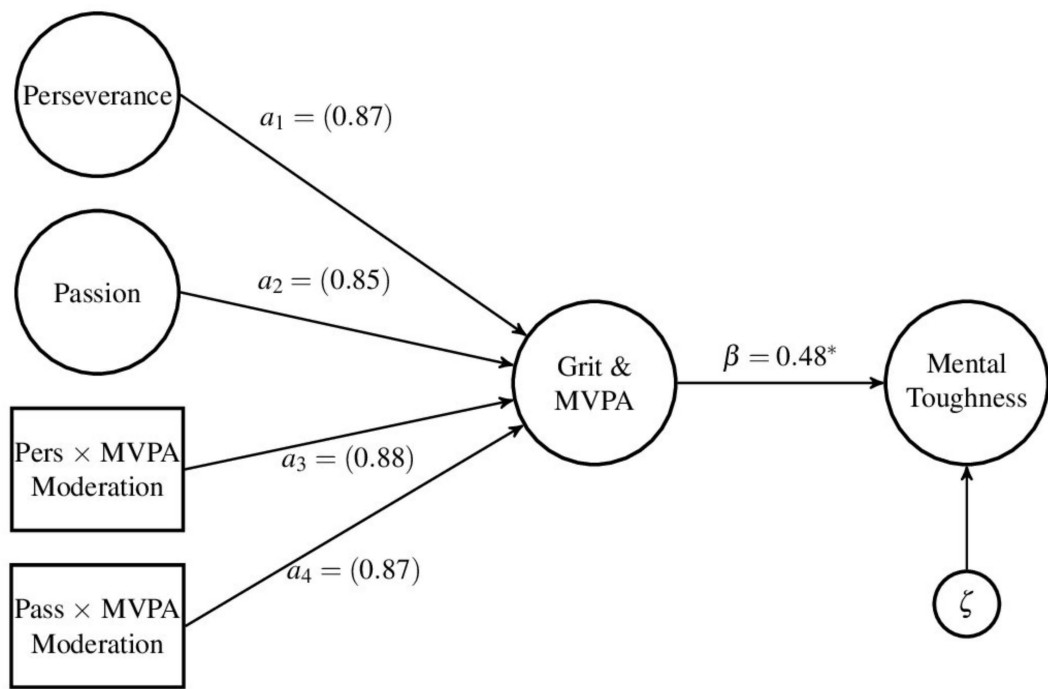

**Figure 3.** Full path model between grit, mental toughness, and moderate-to-vigorous physical activity. * $p < 0.001$.

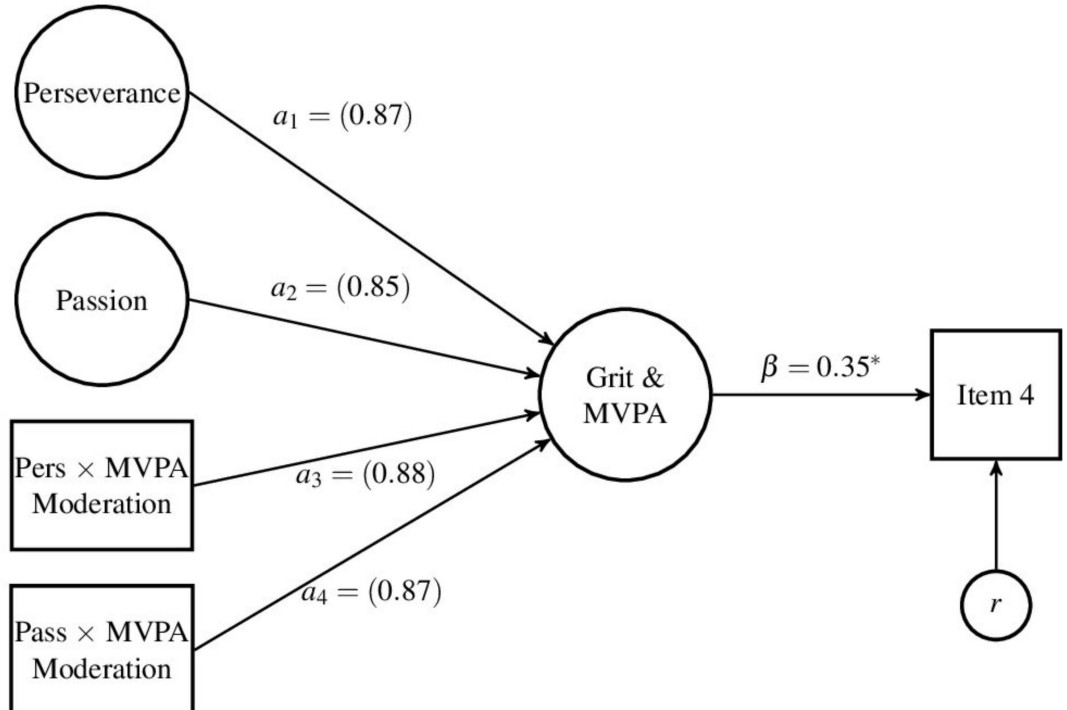

**Figure 4.** Item 4 path model between grit, mental toughness, and moderate-to-vigorous physical activity. * $p < 0.001$.

## 4. Discussion

The objective of this exploratory/hypothesis-generating study encompassed two aspects: The first aim was to conduct a theoretical examination of 2015 Gucciardi's MT conceptual framework within the specific demographic under investigation, addressing RQs 1 and 2. Subsequently, upon establishing the viability of the theoretical framework, we

aimed to empirically test and derive practical implications based on the 11th edition of the ACSM's GETP (RQ 3). While our initial thoughts regarding the RQs were challenged (e.g., due to the uniform adherence of all student-athletes to a minimum requirement of 75 min of MVPA per week, meaningful comparisons between student-athletes and graduate students were unfeasible), we adapted and employed alternative methods to analyze the data as effectively as possible. In this section, we analyze the findings of this study, elucidating their implications in relation to the RQs posed.

Due to the presence of high multicollinearity within the dataset, our initial intention for a step-by-step analysis approach to address each RQ separately was not feasible. Instead, employing PCA and additional regression models allowed us to address all research questions simultaneously. In general, the comprehensive analysis yielded favorable outcomes, providing evidence to answer all three RQs.

More specifically, the results suggest that grit, along with its subcomponents, explains a portion of the variability in MT among US collegiate students. While we are unable to precisely quantify the specific percentage contribution of grit in isolation, the weighted combination of variables, represented by a single weighted component, accounted for 23% of the variability in MT. The regression coefficient indicates a positive relationship between grit and MT, suggesting that higher levels of grit may be associated with higher levels of MT. Direct comparison of our findings with previous research for consistency is not possible as our research explores a novel perspective on the relationship between MT and grit, drawing from a specific MT conceptualization (However, there is previous research that has investigated the relationship between MT and grit, but using a different conceptualization of MT, such as [44]); nonetheless, they support Gucciardi and colleagues' [3] framework that conceptualizes MT as partially explained by grit via *success mindset* (Item 4).

Similar patterns were observed when examining the relationship between grit and US collegiate students' response to Item 4 of the MTI. The grit and MVPA component explained about 12% of the variability in responses to Item 4. The regression coefficient indicates a positive association between the grit and MVPA component and students' responses to Item 4, suggesting that higher levels of grit may be related to a stronger drive for continued success among both samples of US collegiate students. Again, our analysis does not allow for a direct estimation of the isolated contribution of grit in terms of percentage. Nonetheless, the present findings—although they cannot be compared with previous research—provide support for the assertion made by Gucciardi et al. [3] that there is a partial alignment between grit and Item 4.

The above relationships between grit and MT, as well as grit and students' response to Item 4, were examined in the context of MVPA. The results indicate that engaging in at least 75 min of MVPA acts as a moderator, further strengthening the relationship between grit and MT (RQ1) and grit and students' response to Item 4 (RQ2). Specifically, the positive associations between grit and both MT and students' response to Item 4 were amplified when individuals participated in the minimum-by-ACSM MVPA requirement. This suggests that the combination of grit and MVPA may have a synergistic effect on MT and the drive for continued success among US undergraduate student-athletes and graduate students.

In summary, the aforementioned findings suggest a potential avenue for practical application. The positive association between MT and both academic and physical performance underscores the significance of enhancing MT to attain higher performance levels, whether as a student-athlete or graduate student. While grit tends to remain relatively stable over time [52], MT can be developed [53]. *If* the relationship between grit and MT suggests that grit contributes to MT, engaging in at least 75 min of MVPA serves to strengthen this relationship. Notably, MVPA stands as the main modifiable factor between grit and MVPA. Therefore, one practical recommendation for optimizing MT and subsequently enhancing performance among graduate students (the dichotomous variable did not apply to the student-athletes) is to incorporate a minimum of 75 min of MVPA per week, irrespective of individual grit levels. The reported findings from the Council of

Graduate Schools (CGS) indicate a substantial enrollment of a minimum of 1.7 million graduate students in US graduate schools during the Fall of 2021 [54], suggesting a potential for widespread application. However, it is important to note that the recommendation provided is a hypothesis. While one of the general objectives of this research endeavor was hypothesis generation, it is imperative that these hypotheses undergo empirical validation. Several additional recommendations for future research are outlined below.

### 4.1. Future Studies

The existing research on MT and grit has primarily utilized cross-sectional designs [55,56]. Therefore, longitudinal grit–MT research may be needed to avoid the "snapshot" data representation. In addition, our current design/analysis did not allow us to explicitly address the causal direction of the grit–MT relationship. Consequently, multiple measurements would assist not only in mitigating the limitations associated with single-time measurements (e.g., high susceptibility to random sampling error), but also in determining the temporal order and direction of the causal influence between MT and grit. Furthermore, future investigations should explore the impact of different levels of MVPA on the relationship between grit and MT. Specifically, it would be beneficial to examine whether engagement in MVPA below the threshold of 75 min per week significantly affects the grit–MT relationship. Likewise, exploring potential ranges of minutes of MVPA per week (min–max) and discrete categories (e.g., low, medium, high) may provide a more nuanced understanding of the relationship between MVPA, grit, and MT (especially in student-athletes who, on average, appear to satisfy the minimum ACSM recommendation). Lastly, our data collection coincided with COVID-19 health crisis interventions (e.g., remote learning, suspension of collegiate championships); therefore, it would be interesting to examine if similar results could be reproduced post-COVID-19.

### 4.2. Limitations

Our study is not exempt from limitations. Firstly, our findings are based on a convenience sample. While all convenience samples have reduced generalizability compared to probability samples, we argue that homogeneous convenience samples (i.e., US undergraduate student-athletes and graduate students) offer clearer generalizability to those subgroups compared to conventional convenience samples, which are more heterogeneous in nature [57]. Secondly, the design is cross-sectional. Due to the single-time measurement of exposure and outcome, inferring causal relationships becomes challenging. Nevertheless, based on the exploratory/hypothesis-generating approach of our study (e.g., evaluate both exposure and outcome during the course of the same study in a relatively rapid and cost-effective manner and be used as a precursor to planning cohort studies by establishing a baseline), this methodological design was deemed necessary [58]. Thirdly, our data are self-reported. Subjective data are considered limited due to memory bias, social desirability bias, and limited verifiability [59]. At the same time, our research objective was to test Gucciardi's MT conceptualization. Additionally, previous evidence has indicated that individuals may harbor concerns when reporting their levels of MT (perception of weakness or sensitivity if low; [60]). Consequently, the utilization of a self-administered questionnaire (over, for example, an interview), and particularly of the MTI, was considered indispensable to our study design/objectives. Fourthly, we acknowledge that the significant difference in sample sizes between undergraduate athletes (smaller group) and graduate students (larger group) might introduce bias and affect the generalizability of our findings. This discrepancy could lead to an overrepresentation of the experiences and characteristics of the larger group (graduate students) in our results. To address these concerns, we employed statistical methods, such as weighted analyses, to balance the influence of each group on our overall results. This approach helps in adjusting for the unequal group sizes, ensuring that the outcomes reflect the characteristics of both populations fairly. Lastly, it is important to consider the delimitation of using a dichotomous variable concerning the exposure of interest (i.e., MVPA). We acknowledge the significance of studying different

levels of exposure when feasible, as this approach strengthens the plausibility of discovering causal relationships between exposures and outcomes via the evaluation of trends or dose–response relationships [61]. Nevertheless, adhering to the guidelines provided by ACSM, we selected a minimum of 75 min per week as the defined exposure for the exploration purposes of our novel investigation (both an a priori hypothesis and a condition in the analyses), resulting in the creation of two possible exposure categories (i.e., yes/no).

## 5. Conclusions

This study represents a novel investigation that examines Gucciardi et al.'s ([3]) MT conceptualization within the specific subgroups of US undergraduate student-athletes and graduate students, while also incorporating ACSM's latest minimum weekly recommendations for MVPA. Rigorous measures were implemented to ensure a robust level of internal validity. The findings provide initial evidence supporting a positive relationship between grit, engagement in at least 75 min of MVPA per week, and the outcome variables of MT and of Item 4 from the MTI, but they should be approached with caution. Despite the presence of high multicollinearity, which limited the determination of precise unique contributions of the variables, these results may have implications for both basic and applied research, offering valuable insights to researchers and practitioners in the field.

**Author Contributions:** Conceptualization, A.S. and A.B.; methodology, A.S., G.B.M., A.B. and Z.P.; software, A.S., G.B.M., A.B. and Z.P.; validation, A.S., G.B.M., A.B. and Z.P.; formal analysis, G.B.M.; investigation, A.S., G.B.M., A.B. and Z.P.; resources, A.S., A.B. and Z.P.; data curation, A.S., G.B.M., A.B. and Z.P.; writing—original draft preparation, A.S. and G.B.M.; writing—review and editing, A.S., G.B.M., A.B. and Z.P.; visualization, G.B.M.; supervision, A.S.; project administration, A.S., A.B. and Z.P.; funding acquisition, A.S. All authors have read and agreed to the published version of the manuscript.

**Funding:** This research received no external funding.

**Institutional Review Board Statement:** The study was conducted in accordance with the Declaration of Helsinki, and approved by the Institutional Review Board of Clarckson University (protocol number: 21-42; date of approval: 22 February 2021).

**Informed Consent Statement:** Informed consent was obtained from all subjects involved in the study.

**Data Availability Statement:** The data that support the findings of this study are openly available in "OSF" at https://doi.org/10.17605/OSF.IO/NPYT5 (accessed on 8 October 2023).

**Conflicts of Interest:** The authors declare no conflict of interest.

## Appendix A

**Table A1.** Quality Assessment Tool for Observational Cohort and Cross-Sectional Studies.

| Criterion | Yes, Location | No, Comments | Other (CD, NR, NA) Comments |
|---|---|---|---|
| 1. Was the research question or objective in this paper clearly stated? | Introduction | | |
| 2. Was the study population clearly specified and defined? | Participants | | |
| 3. Was the participation rate of eligible persons at least 50%? | | | NA Not a cohort study |

**Table A1.** *Cont.*

| Criterion | Yes, Location | No, Comments | Other (CD, NR, NA) Comments |
|---|---|---|---|
| 4. Were all the subjects selected or recruited from the same or similar populations (including the same time period)? Were inclusion and exclusion criteria for being in the study prespecified and applied uniformly to all participants? | Participants Procedures | | |
| 5. Was a sample size justification, power description, or variance and effect estimates provided? | Data Analysis | | |
| 6. For the analyses in this paper, were the exposure(s) of interest measured prior to the outcome(s) being measured? | | Cross-sectional | |
| 7. Was the timeframe sufficient so that one could reasonably expect to see an association between exposure and outcome if it existed? | | Cross-sectional | |
| 8. For exposures that can vary in amount or level, did the study examine different levels of the exposure as related to the outcome (e.g., categories of exposure, or exposure measured as continuous variable)? | | Dichotomous variable (75 min/week of MVPA: yes/no) | |
| 9. Were the exposure measures (independent variables) clearly defined, valid, reliable, and implemented consistently across all study participants? | Instruments | | |
| 10. Was the exposure(s) assessed more than once over time? | | | NA Cross-sectional |
| 11. Were the outcome measures (dependent variables) clearly defined, valid, reliable, and implemented consistently across all study participants? | Instruments | | |
| 12. Were the outcome assessors blinded to the exposure status of participants? | Data Analysis | | |
| 13. Was loss to follow-up after baseline 20% or less? | | | NA Cross-sectional |
| 14. Were key potential confounding variables measured and adjusted statistically for their impact on the relationship between exposure(s) and outcome(s)? | Data Analysis | | |

Note. In order to make this table more user-friendly, we added the location for when we satisfied a criterion and comments for when we did not; CD = cannot determine; NA = not applicable; NR = not reported.

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
