# Peer review of "The Positive Association between Grit and Mental Toughness, Enhanced by a Minimum of 75 Minutes of Moderate-to-Vigorous Physical Activity, among US Students"

_psych, doi:10.3390/psych6010014_

Round 1

Reviewer 1 Report

Comments and Suggestions for Authors

The work is well presented and certainly interesting. Its current limits are well clarified and explained by the authors. I think it definitely deserves to be published.

Author Response

"The work is well presented and certainly interesting. Its current limits are well clarified and explained by the authors. I think it definitely deserves to be published."

Thank you!

Reviewer 2 Report

Comments and Suggestions for Authors

I would like to thank the editor for the opportunity to review this study and I am flattered to be able to make my contribution. Overall, I consider this article to be well written and a good topic for discussion on the relationship between grit and mental thoughness enhanced by physical activity. However, in my opinion the article could be improved by incorporating some suggestions: 

(a) In the sample section, they could incorporate a table with the most significant data used in the sample description, partially reducing the information in the text.

b) The authors could expand the practical applications on pages 9 and 10 a little, detailing and illustrating the recommendations to a greater extent.

Author Response

Dear Reviewer,

First and foremost, we extend our sincere gratitude for your thoughtful review and constructive feedback on our manuscript. Your insights have undoubtedly enriched our work, and we are grateful for the opportunity to refine our study based on your recommendations.

Regarding your first suggestion about the sample section, we have taken your advice into careful consideration. We are pleased to inform you that we have now included a table summarizing the key data used in the sample description. This addition not only enhances the clarity and accessibility of our demographic and sample-related information but also streamlines the text in the Participants section, as you rightly suggested. We believe this improvement aligns with your recommendation and effectively addresses the concern you raised.

As for your second suggestion concerning the expansion of practical applications in the manuscript, we have given this considerable thought. While we fully appreciate the potential value of elaborating on the practical implications of our findings, we have decided to maintain the current level of detail in this section. This decision stems from our cautious approach to interpreting the results, especially given the high level of multicollinearity observed in our data. Such statistical challenges have necessitated a conservative stance on drawing inferences and making recommendations based on our findings. Our priority is to ensure that any suggestions or practical applications we propose are robustly supported by the data and analysis. In this context, we feel it prudent to refrain from extending this section further, to avoid overstepping the bounds of our data's implications.

We hope you understand our rationale for this decision and trust that it reflects our commitment to scientific rigor and integrity. We are deeply committed to contributing valuable insights to the field, and we believe that our careful interpretation of the results serves the best interests of the research community and the broader audience of our work.

Once again, we thank you for your valuable feedback and for the opportunity to enhance our manuscript. We remain open to further discussion and are eager to engage with any additional thoughts or suggestions you might have.

Respectfully,
the Authors.

Reviewer 3 Report

Comments and Suggestions for Authors

The authors present a prospective study of the complex mind-body relationship as it relates to mental attributes and moderate to vigorous exercise. Using paradigms already in existence, the authors recruited undergraduate student-athletes and compared them to graduate students. A total of 340 participants were involved. The vast majority were female and Caucasian.

The relationships between grit, mental toughness, and at least 75 minutes/week of exercise were assessed with a variety of self-reported responses to validated questionnaires. The statistics are appropriate and standard.

Not surprisingly, student-athletes spent more times exercising compared to graduate students.

The grittier one was or became, the greater the mental toughness. However, it appeared that grit remained relatively stable over time, but the addition of exercise to grit enhanced mental toughness.

I have only a few questions.

Why did the authors use 75 minutes as the cutoff value for exercise? They make this number part of their a priori hypothesis but then present it as a post hoc condition of their analyses. There was tremendous variability in exercise time in both groups.

Was this study supervised by an IRB that was able to allow assessment of students from anyway electronically?

Lastly, could the pandemic with remote learning and other problems at universities have any effect on the results of this study?

Author Response

Dear Reviewer,

Thank you for your insightful comments and questions regarding our manuscript. We would like to address each of your points in turn to provide a comprehensive response.

  1. IRB Supervision and Electronic Assessment of Students: Yes, this study was conducted under the supervision of an Institutional Review Board (IRB) that approved the electronic assessment of students. We adhered to all ethical guidelines and procedures for conducting research involving human subjects. This approval ensured that our data collection methods, including electronic surveys, were ethically sound and compliant with regulatory standards. Details of the IRB approval and ethical considerations will be included in the final version of the manuscript, within a dedicated section outlining author notes and additional study details, to be added upon acceptance of the manuscript.

  2. 75-Minute Cutoff for Exercise: The selection of the 75-minute threshold for weekly moderate-to-vigorous physical activity (MVPA) is based on the American College of Sports Medicine (ACSM) guidelines, which recommend a minimum of 75 minutes of vigorous-intensity aerobic activity per week for adults. This well-established benchmark in sports medicine and physical fitness was integrated into our study design from the outset, serving as a predefined criterion for assessing physical activity levels among participants. Our reference to this cutoff in the analysis was intended to clarify its role as a foundational element of our research framework, rather than a criterion determined after data collection. To eliminate any ambiguity regarding its application, we have explicitly stated in the Limitations section that this cutoff was established a priori and consistently applied throughout our study. This clarification aims to reinforce the methodological rigor and adherence to established physical activity standards in our research approach.

  3. Impact of the COVID-19 Pandemic: We recognize the potential impact of the COVID-19 pandemic on our study results. The pandemic brought significant changes to university life, including remote learning and the suspension of collegiate championships, which could have influenced students' physical activity levels, mental toughness, and grit. In our section on Future Studies, we acknowledge this by stating, "Lastly, our data collection took place during the COVID-19 health crisis (remote learning, suspension of collegiate championships); therefore, it would be interesting to examine if similar results could be re-produced post-Covid-19." This acknowledgment indicates our awareness of the unique context in which the study was conducted and the need for further research to understand these relationships in a post-pandemic setting.

In summary, our study was conducted with IRB oversight, using a methodologically sound approach to assess the role of MVPA, as defined by ACSM guidelines, in the relationship between grit and mental toughness. Additionally, we acknowledge the potential impact of the COVID-19 pandemic on our findings and suggest future research directions to explore these associations further in different contexts.

Thank you, 
the Authors.